# Double Deep Q-Network with a Dual-Agent for Traffic Signal Control

**Jianfeng Gu [1], Yong Fang [1,*], Zhichao Sheng [1] and Peng Wen [2]**

[1]  Shanghai Institute for Advanced Communication and Data Science,
    Key Laboratory of Specialty Fiber Optics and Optical Access Networks,
    Joint International Research Laboratory of Specialty Fiber Optics and Advanced Communication,
    Shanghai University, Shanghai 200444, China; jianfeng@shu.edu.cn (J.G.); zcsheng@shu.edu.cn (Z.S.)
[2]  School of Mechanical and Electrical Engineering, University of Southern Queensland,
    Queensland 4350, Australia; Paul.Wen@usq.edu.au
*  Correspondence: yfang@staff.shu.edu.cn

**Abstract:** Adaptive traffic signal control (ATSC) based on deep reinforcement learning (DRL) has shown promising prospects to reduce traffic congestion. Most existing methods keeping traffic signal phases fixed adopt two agent actions to match a four-phase suffering unstable performance and undesirable operation in a four-phase signalized intersection. In this paper, a Double Deep Q-Network (DDQN) with a dual-agent algorithm is proposed to obtain a stable traffic signal control policy. Specifically, two agents are denoted by two different states and shift the control of green lights to make the phase sequence fixed and control process stable. State representations and reward functions are presented by improving the observability and reducing the leaning difficulty of two agents. To enhance the feasibility and reliability of two agents in the traffic control of the four-phase signalized intersection, a network structure incorporating DDQN is proposed to map states to rewards. Experiments under Simulation of Urban Mobility (SUMO) are carried out, and results show that the proposed traffic signal control algorithm is effective in improving traffic capacity.

**Keywords:** adaptive traffic signal control; deep reinforcement learning; Double Deep Q-Network

## 1. Introduction

Adaptive traffic signal control (ATSC), which automatically adjusts traffic signal timing according to real-time states of intersections, has been shown to be an effective way to reduce traffic congestion. For example, fuzzy logic [1], approximate dynamic programming [2], and genetic algorithm [3] are typical adaptive control methods and have all been successfully applied to current traffic signal control problem. In these models, a traffic signal control policy is established based on recorded traffic flow information or abstract human-crafted features. However, transportation systems have large randomness in reality and traffic flows change remarkably in short term. It is still difficult to possess an expansible and applicable ATSC system employing these models.

As a machine learning framework that attempts to approximate an optimal decision-making policy, deep reinforcement learning (DRL) has shown the great potential of traffic light control in complex transportation systems. Recently, a lot of traffic signal control algorithms based on DRL have been successfully applied in the complex traffic environment. Li et al. [4] first adopted the DRL model to build ATSC systems. They used deep stacked auto-encoders (SAE) to extract vital information from real-time traffic flow and then output Q-value that is corresponding to each traffic signal phase. Mousavi, S.S., et al. [5] compared traffic signal control performances between two kinds of DRL algorithms: the deep policy-gradient and the value-function. The results show that the policy-gradient

outperforms the value-function in the traffic signal control. In [6], Gao, J., et al. implemented a DRL agent of traffic signal control in a more complex traffic environment with a four-phase signalized intersection, and demonstrated that the DRL control method has offered higher performances than the fixed timing method [7] and the longest queue first algorithm [8]. In [9], Deep Q-Network (DQN) [10] was utilized by introducing the coordination algorithm for a scalable approach to controlling large-scale traffic lights. However, most of works built the responsive adaptive traffic signal control system based on the two-phase signalized intersection, or neglected the control of left-turn phase as in [6]. Practically, optimally controlling the operation of four-phase signalized intersection is also significant for urban traffic capacity.

There are mainly two kinds of traffic signal control algorithms based on DRL to solve the real-time traffic signal control in a four-phase signalized intersection. One is the control model which ignored the fixed phasing sequence of traffic signal in real-life intersections [11–14]. It defines each phase as an agent action selected according to the maximal Q-value, which changes randomly. Nevertheless, the fixed phasing sequence of the traffic signal is necessary for keeping traffic regulation orders. The other is to define actions as a binary number that indicates either to extend or to terminate the current green phase and move to the next phase [15,16]. However, the binary action method does not clearly indicate which specific traffic streams the agent can control in a four-phase signalized intersection. It means that the agent holding similar states and actions may receive different rewards. Thus non-stationary distributions occur easily, leading to instability in the training process. It is noted that this binary actions can be applied to traffic light with any number of phases. In this paper, we focus only on the control of traffic signals with four-phase. Apart from aforementioned real-time control methods, the traffic light cycle control algorithm, such as [17], can keep phasing sequence fixed as well. However, this algorithm adjusts time of green lights according to the data from past traffic cycles (a traffic cycle consists of four phase time) and has difficulties to track the variety of traffic flow in real-time.

In this work, to efficiently solve the adaptive traffic signal control problem in a four-phase signalized intersection, we propose a Double Deep Q-network [18] (DDQN) with a dual-agent algorithm. The proposed algorithm incorporates two agents (agent *A* and agent *P*), each of which controls different lanes in every road. To keep the traffic phase sequence fixed, two agents holding two actions to match two-phase traffic signals are denoted by busy or waiting states and shift the control of green lights. For improving the observability and reducing the training difficulty of two agents, we build a reward function and a state representation. The reward function is based on the difference between the number of vehicles that cross the stop line and that are waiting during the last time interval. The states representing vehicle positions are collected from the whole observable environment divided into the same small grids. Further, to enhance the agent feasibility and applicability in the traffic control under the complex transportation system, a network structure which incorporates the DDQN algorithm is introduced. We verify the performance of DDQN with the dual-agent in a four-road intersection established by an open source simulator: Simulation of Urban Mobility (SUMO) [19]. Our numerical results show that DDQN with the dual-agent algorithm is more effective and stable than other traffic signal control approaches in reducing congestion of the four-phase signalized intersection.

Our contributions of the paper include the following. (1) We propose a dual-agent structure to make traffic signal phase sequences fixed and control process stable. (2) State representation and reward function are presented to stabilize training process. (3) A network structure is designed for dual-agent training, so as to efficiently ease congestion and significantly improve traffic capacity.

For a more detailed explanation of our control algorithm, we describe the DRL model in the traffic signal control system and introduce the problem statement in Section 2. In Section 3, we present a dual-agent architecture to formulate the traffic signal control problem in a four-phase signalized intersection and define reinforcement learning components: traffic state, agent action and reward setting. In Section 4, a neural network structure incorporating DDQN and the traffic signal control

algorithm are proposed. Simulation results and discussion are provided in Section 5. Conclusions are given in Section 6.

## 2. Model and Problem Presentation

In this section, we build a four-road intersection environment. Based on this environment, a DRL control model is formulated. To address existing problem under this model and traffic system, we present our control methods.

### 2.1. Deep Reinforcement Learning Traffic Signal Controller

In this paper, we consider a complex intersection scenario, as shown in Figure 1. For each incoming road, the innermost lane named $L3$ is for left turning of vehicles and the other lanes (referred as $L2$, $L1$, and $L0$) are for straight going or right turning of vehicles. As shown in Figure 2, the traffic signal control problem is addressed by a reinforcement learning model, in which the traffic light is regarded as a DRL agent that changes their actions (traffic signal phase) by interacting with the traffic environment. Specifically, the DRL model defines the traffic signal control problem as a Markov decision process (MDP), which mainly contains five elements $<\mathcal{S}, \mathcal{A}, \mathcal{P}, \mathcal{R}, \gamma>$. As usual, $\mathcal{S}$ and $\mathcal{A}$ are spaces of states and actions separately, $\mathcal{P}$ is transition function, $\mathcal{R}$ is known as reward function and $\gamma \in [0,1]$ is the discount factor. At each time step $t$, the agent obtains the state $s_t \in \mathcal{S}$ and chooses an action $a_t \in \mathcal{A}$ based on the observation. In return, the agent perceives the next state $s_{t+1} \sim \mathcal{P}(s_t, a_t)$ and acquires a reward $r_t \sim \mathcal{R}(s_t, a_t)$. This process continues until the agent reaches a terminal state, which is the last observation state in our traffic simulation.

The goal of an agent is to find an optimal policy $\pi : \mathcal{S} \times \mathcal{A} \rightarrow [0,1]$ which maximizes the expected return from each state $s_t$. The expected return $R_t$ at time $t$ is defined as follows:

$$R_t = E[\sum_{k=0}^{\infty} \gamma^k r_{t+k}] \tag{1}$$

where $\gamma$ is the discount factor to acquire a tradeoff between the current and future reward. In the Q-learning algorithm, after an action is selected by $\epsilon$-greedy policy, the expected return becomes

$$Q^\pi(s,a) = E[R_t | s_t = s, a] \tag{2}$$

where $Q^\pi(s,a)$ is the action-value. To obtain the maximal action-value for a state $s$ and an action $a$ achievable by any policy, the optimal value function is defined as

$$Q^*(s,a) = \max_\pi Q^\pi(s,a) \tag{3}$$

Generally, the action-value function in value-based model-free reinforcement learning methods is described by a function approximator, such as a deep convolutional network. Specifically, $Q(s,a;\theta)$ is represented as approximator with parameters $\theta$, and needs to approach to the optimal action value, i.e.,

$$Q^*(s,a) \approx Q(s,a;\theta) \tag{4}$$

where parameters $\theta$ are learned by iteratively minimizing the loss function

$$L(\theta) = E(r + \gamma \max_{a'} Q(s',a';\theta) - Q(s,a;\theta))^2 \tag{5}$$

where $r + \gamma \max_{a'} Q(s',a';\theta)$ is the target value, $s'$ and $a'$ is the state and action at the next time step respectively. In the DQN algorithm, a target network with weights $\theta^-$ is adopted to address the

instability problem of the policy. Thus, the optimal target value is replaced by the approximate target value

$$y = r + \gamma \max_{a'} Q(s', a'; \theta_i^-) \tag{6}$$

where $\theta_i^-$ are parameters of the target network in the iteration $i$. This lead to a sequence of loss function $L_i(\theta_i)$ that changes at each iteration $i$,

$$L_i(\theta_i) \approx (r + \max_{a'} Q(s', a'; \theta_i^-) - Q(s, a; \theta_i))^2 \tag{7}$$

Note that the agent based on the DQN algorithm samples random minibatch of transitions $(s, a, r, s')$ from the experience replay to keep networks converging steadily.

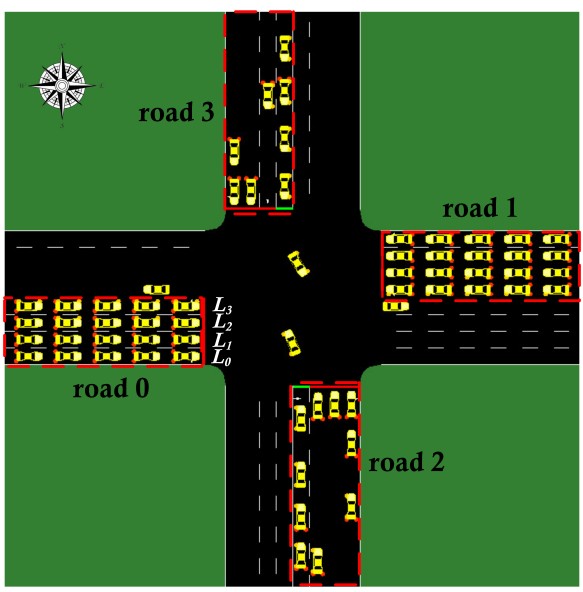

**Figure 1.** A four-road intersection. The red-dashed region represents incoming road.

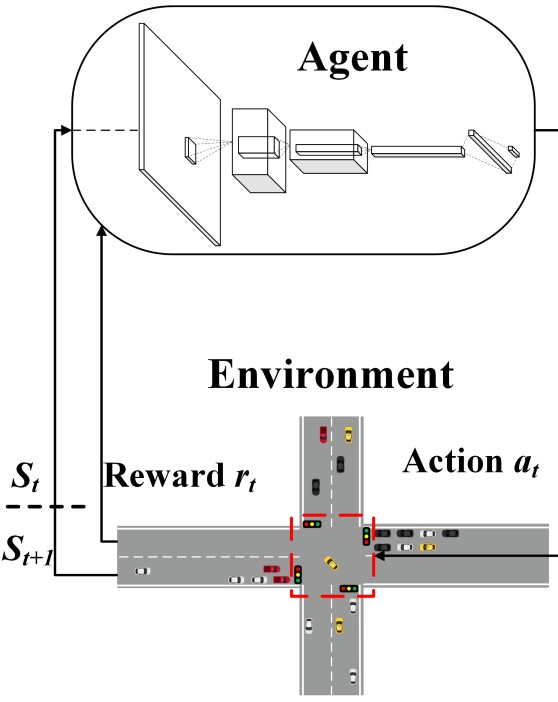

**Figure 2.** The traffic signal controller based on DRL. There is the traffic signal control system referred as agent in the red-dashed region.

## *2.2. Problem Presentation*

ATSC based on DRL methods under a four-phase signalized intersection is a difficult task. On the one hand, the traffic signal may be changed in a random sequence. It is because agent actions that match each phase of a traffic signal are chosen according to the maximal Q-value, which does not change in a specific sequence. On the other hand, it is hard for the information of states and rewards to be extracted accurately and adequately by the traffic signal control agent.

In this paper, we target controlling a traffic signal in a four-phase signalized intersection effectively. Problems lie in how to keep the phasing sequence fixed and extract information from intersections adequately. For the first problem, our approach adopts a dual-agent (i.e., agent *P* and agent *A*) architecture, in which each agent possesses two actions to match two traffic phases. To keep the phase sequence of traffic signals fixed, each agent denoted by busy or waiting state shifts the control of green signals (more details can be seen in Section 3). Due to each action matching each phase, non-stationary transitions that easily appear in the algorithm based on binary actions are avoided. Therefore, the DRL agent in our environment learns an effective control policy faster. For the second problem, we present a reward function and a state representation. The state is represented by a matrix of vehicle positions extracted from the observation area, which is divided into small square-shaped grids according to [17]. It can provide much critical information for agents, such as distances between vehicles and the stop line. The reward function is based on the difference between the number of vehicles that pass through the intersection during the last time interval and those that are staying. All used information is easily obtained from the traffic signal simulator (e.g., SUMO), which is based on a vehicular network [20,21]. Further, to improve the expansibility and stability of two agents in the traffic signal control, we build a deep neural network to map states to rewards.

## 3. Deep Reinforcement Learning Model with Dual-Agent

In this section, the dual-agent structure is proposed to obtain an effective control algorithm ensuring the fixed phase sequence in traffic signals. The matrix state of vehicle positions and reward functions are presented to enhance the observation ability and reduce the learning difficulty of two agents.

### *3.1. Dual-Agent Architecture*

In the predetermined intersection scenario, one phase corresponds to a green signal state. Thus, considering yellow signal states, a traffic signal has eight control states as shown in Table 1. The road 0 and road 2 are not illustrated because of the symmetry in the intersection. The symbols G, r, y, and g represent the green light, red light, yellow light, and left-turn waiting light, respectively. For example, the state 'GGGg' of road 1 represents that lane states from *L*0 to *L*2 are 'GGG' and the lane *L*3 state is 'g'. It means that the straight traffic flow of road 1 can pass through the intersection, and the left-turn traffic flow of road 1 needs to let the straight traffic flow of road 0 go first and then seek a chance to turn left.

**Table 1.** Traffic signal state.

| Road 1 | Road 3 |
|--------|--------|
| GGGg | rrrr |
| yyyg | rrrr |
| rrrG | rrrr |
| rrry | rrrr |
| rrrr | GGGg |
| rrrr | yyyg |
| rrrr | rrrG |
| rrrr | rrry |

Previous action settings [16] ensure q fixed the traffic signal phase sequence, but easily result in non-stationary action-states distributions that make it difficult to train an agent. Aiming at accelerating agent learning and improving agent performance, we present the dual-agent architecture, where the agent $P$ controls the straight traffic flow (i.e., traffic flow in $L0$, $L1$, and $L2$ of each incoming road) and the agent $A$ controls the left-turn traffic flow (i.e., traffic flow in $L3$ of each incoming road). Specifically, the agent $P$ has two actions (0 and 1): turning on green lights for traffic streams of north–south (NS, i.e., $a = 0$) and traffic streams of east–west (EW, i.e., $a = 1$). Correspondingly, action 0 and 1 of the agent $A$ indicates to give way to left-turn traffic streams of north–south (referred as NSL, i.e., $a = 0$) and east–west (EWL, i.e., $a = 1$), respectively. Note that four phases of traffic signals represent green signal states of NS, NSL, EW, and EWL, which change in a specific sequence.

Figure 3 shows the control strategy of two agents implemented in road 2. At the time step $t_p$, the busy agent (e.g., the agent $P$ denoted by the busy state, the agent $A$ denoted by the waiting state) perceives the traffic state and selects an action 0. Then, the traffic signal state of road 2 switches to 'GGGg' and keeps $\tau_p$ time. There is $\tau_p$ time between two time steps if the transition is not taken into account. Note that road 0 is in the same signal state with road 1, and road 2 is in the same signal state with road 3. At the time step $t_p + 1$, the agent $P$ controls the traffic signal in the same manner. The traffic signal state remains the same until the action of the agent $P$ changes. At time step $t_p + 2$, the busy agent selects the other action 1. The traffic signal state then changes to the transition state 'yyyg' and lasts for $\tau_y$ time. Following the same control method as agent $P$, agent $A$ exchanges the state with agent $P$ and obtains the access to control the green signal after the transition finishes. At the time step $t_a + 2$, the action of agent $A$ changes and the traffic signal state shifts to a transition again. After the second transition is over, agent $P$ denoted by the busy state immediately executes the action 1 and does not receive the traffic state. Therefore, the state directly changes to 'rrrr' and hold for $\tau_p$ time before $t_p + 3$. Significantly, two agents regard the action change as a sign to shift the access of the traffic signal control in the dual-agent architecture.

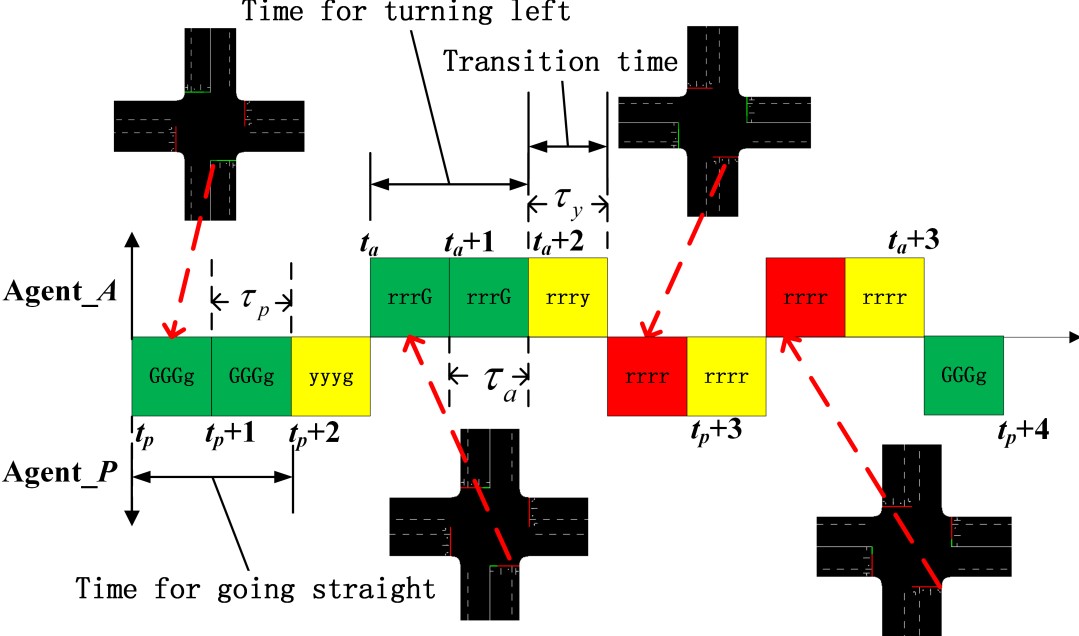

**Figure 3.** The dual-agent control strategy implemented in the road 2. In the time axis (the horizontal axis), the green, red, and yellow rectangle indicates the green, red, yellow signal in road 2, respectively.

The more detailed operation that ensures fixed phasing sequence is illustrated in Figure 4. Agent $P$ and agent $A$ are denoted by two different states (busy and waiting), respectively. It means that only one agent can control the traffic green signal at each time. Agent $P$ controls the traffic signal with phase

0 and phase 2, and agent *A* controls the traffic signal with phase 1 and phase 3. If the agent denoted by busy state changes its action, two agents need to shift their states. Then the traffic signal enters into the next phase. It is because of this operating mode that the proposed dual-agent architecture ensures the fixed phase sequence.

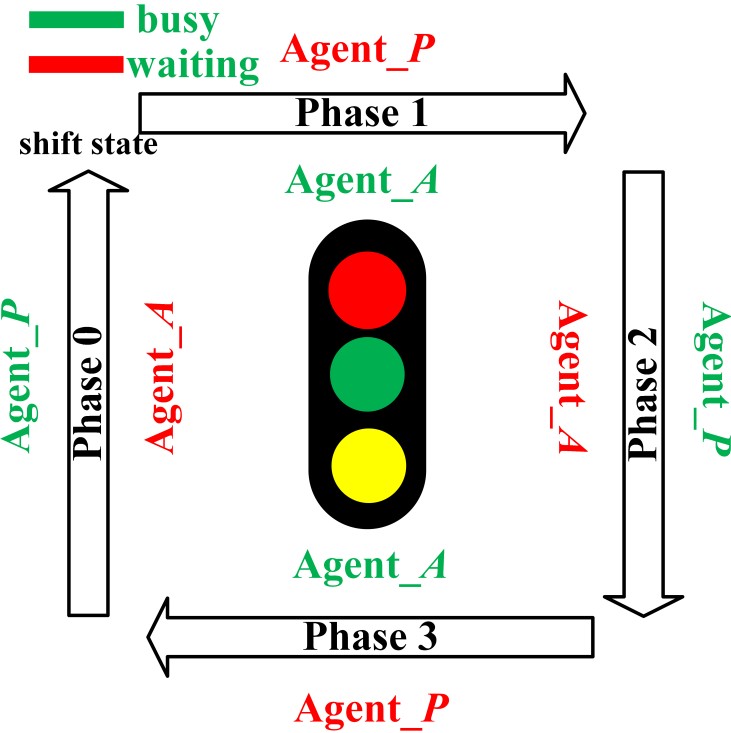

**Figure 4.** Operating mode to ensure the fixed phase sequence. The red and green color indicates waiting and busy state respectively.

*3.2. State Representation*

Taking into consideration the difficulty in real traffic information acquisition, the method of defining the vehicles' speed as part of a state in [6,17] is discarded in this paper. Instead, we define the state based on the position of vehicles at an intersection. Through vehicle detection technology [22], or other tools such as induction-loop and vehicular network, information of vehicle positions can be easily collected. In addition, vehicle positions reflect the traffic environment adequately and offer some detailed information, such as the queue length. The open source simulator SUMO provides the aforementioned traffic information. In the preset intersection environment, each observable incoming lane is divided into same-size small cells. According to the vehicle length and the minimum safe distance, the cell length is set to be *c*. The length of observable lanes starting from the stop line is set to be *l*. The state of each road is a matrix with a binary number. If the midpoint of a vehicle fills the cell, the corresponding coordinate of a matrix is set to be 1, else 0.

For example, Figure 5 shows a state matrix at incoming road 2. The observation area of road 2, starting from stop line, is divided into the same discrete cells with length *c*. Lanes from *L*0 to *L*3 represent rows of state matrix. At each time step, we obtain four state matrices from four incoming roads. Therefore, the agent state is represented by four vertically concatenated matrices, each value of which represents the relative vehicle position in each incoming lane.

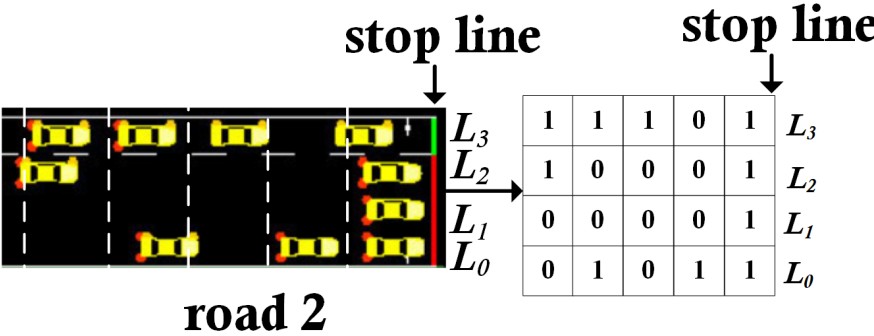

**Figure 5.** The intersection state defined as a matrix. All states are extracted from incoming roads.

*3.3. Reward Function*

For both types of traffic signal controllers in dual-agent architecture, a reward provides a feedback to an intersection agent after executing a given action. Thus, an appropriate reward definition will accurately guide the agent to learn an optimal policy that can effectively ease traffic congestion.

The commonly used reward function is based on the difference of previous and current cumulative delay [5,14]. However, it is hard to acquire cumulative delay of vehicles in a real traffic system because this needs to identify each vehicle and precisely calculate its waiting time. Therefore, for simplicity, we propose two new reward functions for two agents according to the difference between the number of vehicles that pass through an intersection during the last time interval and those that are staying. Specifically, the agent $P$ obtains the number of staying vehicles twice between two time steps to calculate the difference. Agent $P$ reward function at time step $t$ is defined as

$$r_{pt} = O_{pt} - PS_t \qquad (8)$$

where $O_{pt}$ is the number of going straight vehicles passing through the stop line during $\tau_p$ time, $PS_t$ is the number of staying vehicles in the other lanes with traffic signal state 'r'. $PS_t$ is regarded as a penalty for agent $P$ to avoid vehicles in other control lanes of the agent $P$ waiting too long. Similarly, the agent $A$ reward function is defined as

$$r_{at} = O_{at} - AS_t \qquad (9)$$

where these parameters are similar to the agent $P$. Unlike the agent $P$, the agent $A$ controls the left-turn green light of each road. It is noted that the information of rewards and agent states is all extracted from incoming lanes.

## 4. Double Deep Q-Network with Dual-Agent Algorithm

In this section, we propose a network structure with a dual-agent on the basis of DQN. The new structure incorporates DDQN to handle the over-estimation of Q-value. Then, we present the traffic signal control algorithm.

*4.1. Neural Network Structure*

Inspired by networks of DQN, we propose a network structure to train the agent $P$ and agent $A$. The whole network structure for two agents is shown in Figure 6. To obtain an effective traffic signal control policy, convolutional neural networks (CNNs) are used to extract critical information from vehicle position matrix. Parameters of convolutional networks are shared between two agents since the traffic environment is public.

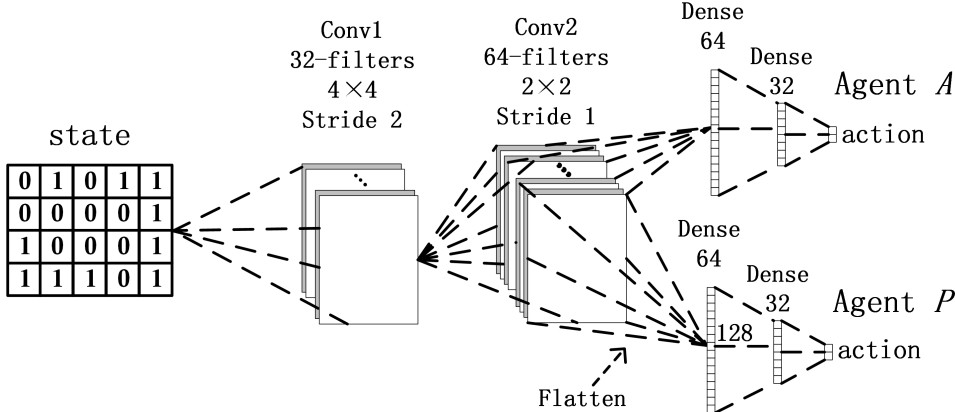

**Figure 6.** The network structure of the dual-agent architecture.

The state is fed into the first convolutional layer with 32 filters of 4 × 4 with the stride 2 and the second convolutional layers with 64 filters of 2 × 2 with stride 1, where both of them leverage the Rectified Linear Unit (ReLU) as the activation function. Finally, the fully-connected networks (belonging to busy agent) with 64 (ReLU) and 32 hidden units linearly generate Q-values whose dimensions represent agent actions.

To prevent the overestimation of the Q-value, the DDQN algorithm is adopted. It defines a new target as

$$Q_{target} = r + \gamma Q(s', \arg\max_{a'} Q(s', a'; \theta_i)) \tag{10}$$

where $\theta_i$ is the evaluating network parameters at the iteration $i$. In (10), the maximized Q-value of the DQN target is replaced by an action $a'$ based on the maximal Q-value of evaluating networks to indirectly reflect the best Q-value. Therefore, the overestimation of the Q-value is eliminated.

### 4.2. Traffic Signal Control Algorithm

To effectively control the traffic signal, we propose a DDQN with a dual-agent algorithm. Its pseudocode is indicated in Algorithm 1. Two agents have the same initial action, which keep the traffic phase sequence fixed. At each time step, the busy agent (e.g., the agent $P$ first) perceives the traffic state, and adopts the $\epsilon$-greedy algorithm to select an action $a_p$. After the action $a_p$ is implemented, traffic states change from $s_p$ to $s'_p$. Then, it obtains a reward $r_p$ calculated by the (8). All this information forms a record (or a transition) $(s_p, a_p, r_p, s'_p)$ that need to be put into experience $MP$. At the end of the time step, the agent $P$ randomly selects mini-batch data from the experience $MP$ to train, then based on (11) which is shown in the follow to update evaluating network parameters $\theta$.

$$\theta_i \leftarrow \theta_{i-1} + r_l(y_i - Q(s, a; \theta_i)) \nabla \theta_i Q(s, a; \theta_i) \tag{11}$$

where $a$ is agent action, $r_l$ is the learning rate. At every training iteration, the agent $P$ updates parameters $\theta^-$ of the target network based on:

$$\theta^- = (1 - \beta)\theta^- + \beta\theta \tag{12}$$

where $\beta$ is the target network update rate. The agent $A$ has the same training process as agent $P$ except for the experience memory, parameters $\epsilon$, and reward functions.

---

**Algorithm 1** Double Deep Q-Network with Double-Agent for Traffic Light Control

---

 1: Initialize network parameters, $\theta$ with random values
 2: Initialize two agent replay memory $MP$ and $MA$ with capacity $L$
 3: Initialize $N,T,\tau_p,\tau_a$
 4: **for** episode = 1,2,...,$N$ **do**

 5:     initialize agent $P$ historical states $s_p$ and agent $A$ historical states $s_a$
 6:     start a new time step and set the agent $P$ state to busy
 7:     **for** $t = 0$ to $T$ - 1 s **do**

 8:         **if** new time step begins **then**

 9:             busy agent chooses an action ($a_p$ or $a_a$) according to $\epsilon$-greedy policy
10:             **if** action changes **then**

11:                 execute the transition
12:             **end if**
13:         **end if**
14:         **if** transition ends **then**

15:             exchange two agent states
16:             directly execute the busy agent selected action
17:         **end if**
18:         add a second, $t = t + 1$
19:         **if** busy agent (agent $P$ or $A$) time step ends **then**

20:             busy agent observes a new state ($s'_p$ or $s'_a$)
21:             busy agent acquires a reward $r$
22:             busy agent places a record $(s,a,r,s')$ in memory
23:             $s_p \leftarrow s'_p$ or $s_a \leftarrow s'_a$
24:             busy agent training
25:             start a new time step
26:         **end if**
27:     **end for**
28: **end for**

---

## 5. Results and Discussion

In this section, simulation experiments are conducted to evaluate the performance of the DDQN with dual-agent algorithm. First, the simulation setup and parameters are presented. Then we set neural network hyperparameters. Finally, traffic control simulations are implemented to validate the performance of DDQN with dual-agent algorithm in term of improving traffic capacity.

### 5.1. Simulation Setup and Parameters

The open source simulator SUMO (version 1.3.1 in this paper) based on the future vehicular network provides necessary vehicular information, such as positions and waiting time, which helps the agent calculate rewards and extract states. In this paper, traffic environment settings follow the research [6]. The intersection is composed of four roads, as shown in Figure 1. Each incoming road has four lanes, where the right-most lane ($L0$) is for straight and right-turn traffic flow, two middle lanes ($L1$, $L2$) allow vehicles to go straight only, and the left inner lane ($L3$) allows vehicles to turn left.

The method by which vehicles are generated and released into the network has a major impact on the quality of any traffic simulation. The most popular vehicle production method is to randomly sample from a probability distribution numbers that respect to the time interval between vehicles. This work does not break from this method totally, however we struggle for implementing a nuanced version which better models realistic scenarios. Empirical research has demonstrated that different vehicle flow rates are suitably approximated by different probability distributions [23]. In this paper, the vehicle production rate follows the Bernoulli process, and the probability $p$ (referred as production rate) setting can be seen in Table 2. The roads of east-west (direct routes), for example, have 1/6 vehicle production probability for the road 0 and road 1 respectively. The sum of production rates of all roads

is 1, which represents that one vehicle enters into the intersection every second. It is obvious that the traffic flow density is 3600 veh/h in the intersection.

**Table 2.** The production rate on different directions.

| Traffic Stream | Production Rate |
|---|---|
| Straight traffic flow of east-west | 1/6 |
| Left traffic flow of east-west | 2/15 |
| Straight traffic flow of north-south | 1/10 |
| Left traffic flow of north-south | 1/10 |

To simulate the intersection traffic flow and traffic signal control, it is significant to set appropriate traffic environment parameters. In this work, the simulated intersection is a 500 m $\times$ 500 m area, and each observable road (from the stop line) length $l$ is set to be 160 m. In the SUMO simulation environment, the vehicle length is 5 m and the minimum safety distance is 2.5 m. Therefore the cell length $c$ is set to be 8 m. Road speed limit is set to be 70 km/h. The time intervals $\tau_p$, $\tau_y$, $\tau_a$ are all set to be 6 s. This interval setting gives enough time for driver to react and alleviates pressures of the real-time control in traffic system.

To acquire an effective traffic signal control policy, the DDQN with dual-agent algorithm is trained for 500 episodes, each of which contains 1800 s. Detailed parameter settings of dual-agent networks are illustrated in Table 3. In the network training process, we employ the ADAptive Moment estimation (Adam) [24] that updates the learning rate considering both first-order and second-order moment with the stochastic gradient descent procedure. The learning rate is set to be 0.0002.

**Table 3.** Parameters in Double Deep Q-Network with dual-agent.

| Parameter | Value |
|---|---|
| Replay memory size of $MP$ or $MA$ | 10,000 |
| Minibatch size $b$ | 32 |
| Starting $\epsilon$ | 1 |
| Ending $\epsilon$ | 0.05 |
| Reduction per time of $\epsilon$ | 0.0005 |
| Target network update rate $\beta$ | 0.001 |
| Learning rate $r_l$ | 0.0002 |
| Discount factor $\gamma$ | 0.9 |

*5.2. Experimental Results*

As illustrated in [25], there are mainly three standard action definitions, phase switch [16], phase duration [17], and phase itself [11]. In this paper, we focus on the fixed phasing sequence in traffic signal control. Therefore, the performances of a phase control method that ignores the fixed phasing sequence are not discussed in our work. The control algorithm of phase duration (i.e., the traffic light cycle control algorithm) has difficulties to track the variety of the traffic flow in real-time because there is at least one traffic light cycle time between two time steps. The big gap between two time steps often makes the traffic signal control a more Partially Observable Markov Decision Process (POMDP) [26], leading to large variance in training results. Thus, the method with phase duration is not compared with the proposed algorithm as well.

To verify the performance of the proposed DDQN with a dual-agent algorithm, we make comparisons of training results among the proposed algorithm (referred as DDQN with dual-agent), the phase switch (referred to as action with binary number), and DQN algorithm [6](referred as DQN with single agent). The setting of DQN algorithm is the same as [6] because it established the control architecture in the same intersection scenario. The control algorithm with the binary action has similar settings as our algorithm except that it has one agent setting for the sake of comparison. We use one

component (e.g., agent *P*) of the dual-agent network to train the single agent. In addition, we find that the network of the control algorithm with binary action is not convergent in the predetermined intersection scenario when utilizing the reward function based on the difference of traffic flow between two time steps. Therefore, a modified reward function based on the difference of previous and current cumulative vehicles delay is used to ensure the convergence stability. Note that these algorithms adopt the same classical DQN network structure, which has little influence on their compared results.

Simulation results of the training are summarized in Figures 7 and 8, in which the average waiting time and the number of arrived vehicles of each episode are presented, respectively. Specifically, average waiting time is the average value of vehicle delay, and the number of arrived vehicles is the sum of vehicles that reach their destination. From Figures 7 and 8, we observe that DDQN with a dual-agent algorithm outperforms the binary action algorithm and DQN algorithm with a large margin. It clearly demonstrates that the agents in the proposed algorithm learn a more effective control policy than other control algorithms. As expected, the control algorithm with binary action easily leads to non-stationary distributions of action-states, which degrades the training performance. In the DQN algorithm, the action definition is similar to phase. However in order to maintain phase sequence fixed, the agent just holds two actions to control direct routes of traffic flow. Thus, the DQN algorithm suffers from the traffic flow of left turns. By contrast, in our DDQN with dual-agent algorithm, one action is distinctly matched to each phase to guarantee stable action-state distributions and achieve a more effective performance.

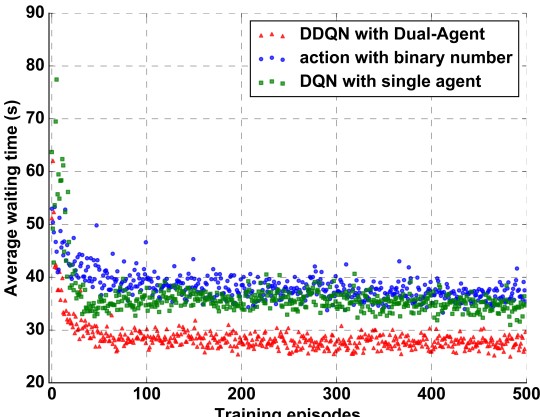

**Figure 7.** Average waiting time when the dual-agent control method, DQN algorithm, and control algorithm with binary action are used.

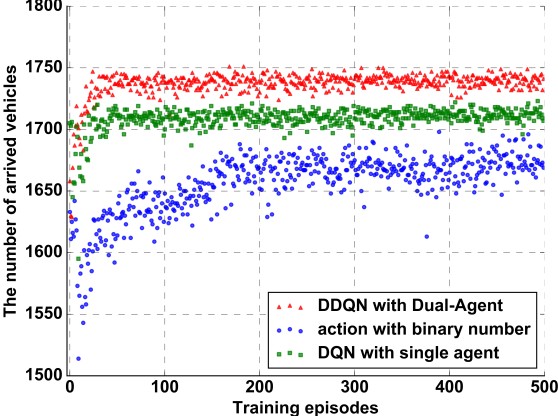

**Figure 8.** The number of arrived vehicles when the dual-agent control method, DQN algorithm, and control algorithm with binary action are used.

To validate the expansibility and stability of the DDQN with dual-agent algorithm, we make a comparison among our control algorithm, the fixed timing method [7], the DQN algorithm that does not consider left-turns [6], and control algorithm with binary action [16]. Following the same traffic production rate in Table 2, we change the traffic production by a ratio *r* (to multiply the production rate). In order to simulate the low, medium, and high traffic volume, the ratio *r* values are set to be 0.5, 0.7, 0.9, 1.0, 1.1, and 1.2. Correspondingly, the traffic production is 1800, 2520, 3240, 3600, 3960 and 4320 veh/h respectively. For each value in traffic production, we conduct 50 evaluating experiments (3600 s per experiment, the same production rate as shown in Table 2) with different random seeds for each control algorithm. Note that these DRL control methods are trained in the traffic system with 3600 veh/h traffic production. We calculate average values from these experiments as our results, which are illustrated from Figure 9. It is noted that the DDQN with dual-agent algorithm has an effective performance on three aspects, average waiting time, vehicle stop times, and the number of arrived vehicles. Due to the adequate capacity in the four-road intersection at low vehicle production rate, there are no significant differences in the number of arrived vehicles among these algorithms in Figure 9c. However, when the available capacity for vehicles is degrading along with the increasement of traffic productions, the DDQN with dual-agent algorithm provides the better performance than other control algorithms in terms of the number of arrived vehicles. Furthermore, comparing to control algorithm with binary action, our control algorithm has a more stable and effective control ability in most cases, especially in high traffic productions. This poor performance is caused by traffic congestion. The Figure 10 (described hereafter) illustrates the sum of vehicles existing in an intersection per second. As shown in the results under the high traffic production (4320 veh/h), the vehicle number in the phase switch algorithm continues to increase, resulting in traffic congestion. On the contrary, the proposed algorithm ensures sustainable vehicle number and preserves unimpeded traffic flow.

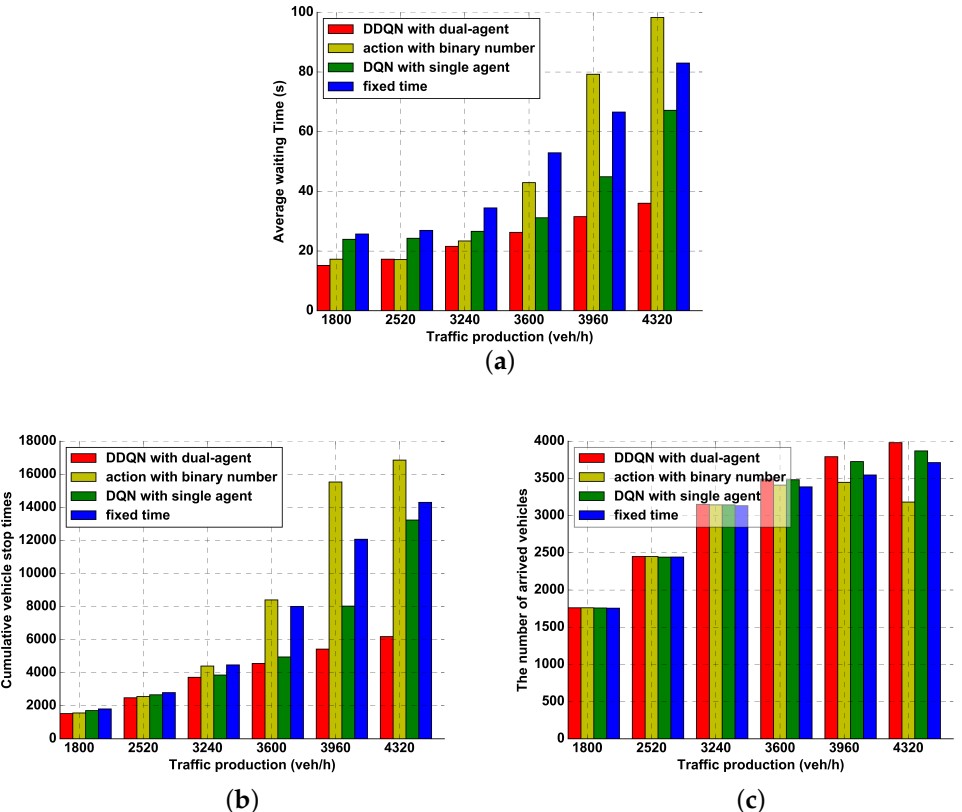

**Figure 9.** Comparison performances of different traffic signal control algorithms: (**a**) average waiting time; (**b**) the number of stops; (**c**) the number of arrived vehicles.

For testing whether the intersection suffers from congestions, we compare the sum of vehicles that exist in an intersection for every second with different algorithms too. It is noted that these control methods are also trained in the traffic environment with 3600 veh/h traffic production (i.e., ratio $r = 1$). Under the same traffic production rate in Table 2, the vehicle productions are set to be 2520 veh/h, 3600 veh/h and 4320 veh/h (i.e., ratio $r$ =0.7, 1.0, and 1.2 separately) for each comparison, which contains 50 test experiments. Figure 10 plots the test curve of each control method, where the solid shows the average existing vehicles per test episode and the shade shows its standard deviation. Since the traffic volume is hard to be saturated under the low traffic production, there are no significant differences among these methods (see Figure 10a). In Figure 10b,c, due to the non-stationary distribution, the phase switch (binary number) method is failed to maintain sustainable vehicle number. As expected, the performance of DQN algorithm is limited by traffic volume of left-turn, particularly in the high traffic production. On the other hand, the proposed algorithm shows the best and the most robust control ability as its test curve keeps stable and has small standard deviation.

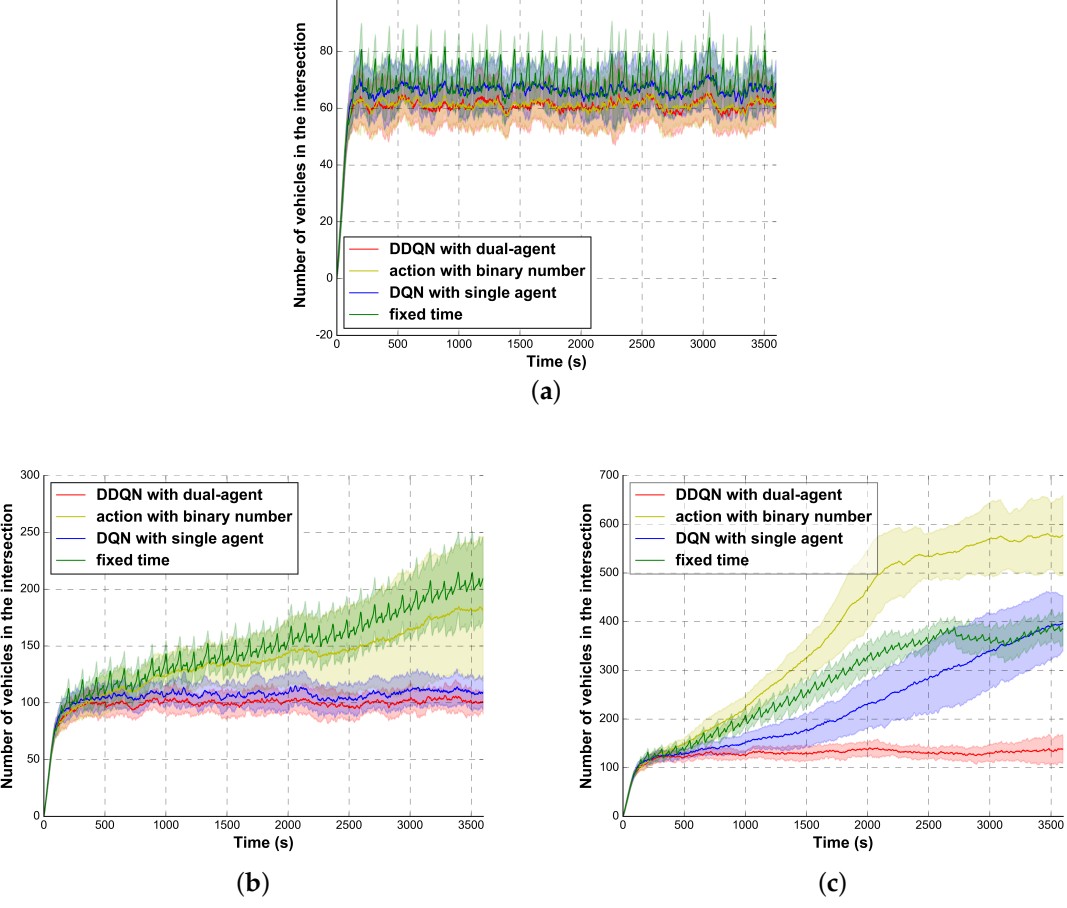

**Figure 10.** The number of vehicles existing in the intersection per second: (**a**) vehicle production 2520 veh/h; (**b**) vehicle production 3600 veh/h; (**c**) vehicle production 4320 veh/h.

Further, to verify the feasibility and practicality of the DDQN with dual-agent algorithm, we make a comparison (the same setting as above experiment) in improving the efficiency of the intersection between our algorithm and actuated traffic lights based on time gaps. Its mechanism is to prolong a traffic phase whenever a continuous traffic stream is detected and changes to the next phase after detecting a sufficient time gap between successive vehicles. In this paper, the maximum time gap (i.e., sufficient time gap) is set to be 2 s, which is found to be the optimal value in our simulation environment. To be fair, minimum phase time in the actuated traffic light and yellow light time are

set to be 6 s that is the same as a DDQN with dual-agent algorithm. The comparison results (the ratio $r$ from 0.5 to 1.2) are shown in Figure 11. The results indicate our algorithm is more efficient to resolve traffic congestion in most cases except that traffic productions is 1800, 2160, 2520, and 2880 veh/h, respectively.

There are two possible reasons for this performance. The first is that the neural network is trained in the traffic system with 3600 veh/h traffic production. The second is that the proposed algorithm uses a longer time interval ($\tau_p$, $\tau_a$) than actuated traffic light. To further prove our viewpoints, we re-train the DRL agent in two different setting. The first is to set the ratio $r = 0.7$ (i.e., $3600 \times 0.7 = 2520$ veh/h traffic production), $\tau_p = 6$ s, and $\tau_a = 6$ s (referred as $r = 0.7\ \tau_p = 6\ \tau_a = 6$), the second is to set the ratio $r = 0.7$, $\tau_p = 3$ s, and $\tau_a = 3$ s (referred as $r = 0.7\ \tau_p = 3\ \tau_a = 3$). Test results (other setting is in the same before) are shown in Figure 12. Compared to the former test results, the performances under the two settings have indeed improved a little. Therefore, for both reasons, generalization ability of neural network and the time interval, have impacts on the performance of DDQN with dual-agent algorithm when in low traffic production.

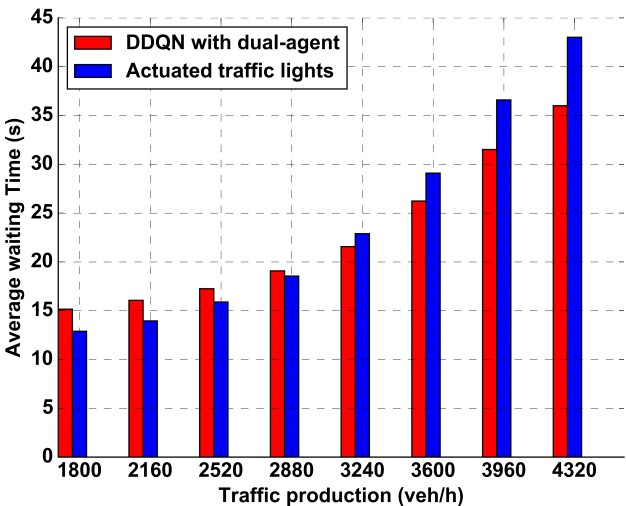

**Figure 11.** Comparison performance of average waiting time between DDQN with dual-agent and actuated traffic signals.

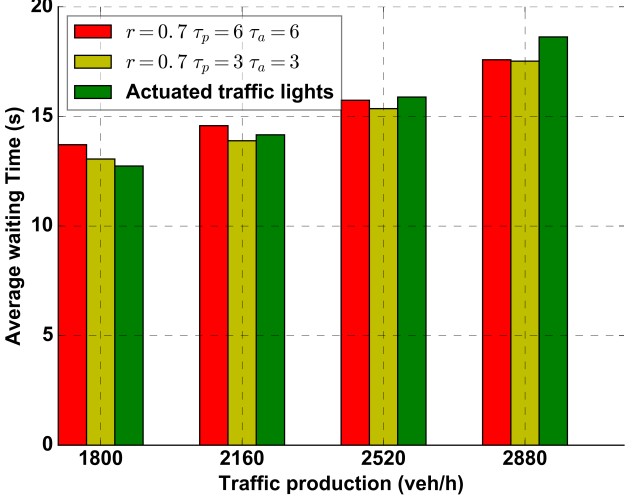

**Figure 12.** Comparison performance of average waiting time between DDQN with dual-agent and actuated traffic signals under different ratio $r$ and control time interval.

## 6. Conclusions and Discussion

In this paper, a DDQN with dual-agent algorithm is proposed to effectively reduce congestion at a four-phase signalized intersection. The proposed architecture includes two agents, agent *P* and agent *A*, where each agent controls traffic flow with different directions. In the traffic signal control process, two agents shift behaviors from working and waiting. Through the dual-agent control algorithm, the fixed phasing sequence which is commonly ignored in some DRL methods [11,13] is guaranteed. Due to enough intervals between each control action, the traffic light flicker is avoided, and the system computation load is alleviated. The fixed phasing sequence of traffic signal balances waiting time of vehicles for each road as well. Owing to the difficulty in obtaining data from a real traffic system, a reward function based on traffic flow and a state representation with vehicle positions are presented to accelerate the agent learning speed. The algorithm is evaluated in simulations, and results show the performance of the dual-agent algorithm in improving traffic capacity is much better than the binary action control algorithm.

To coordinate two agents, we set a penalty factor in reward functions in previous experiments. It is used to avoid vehicles waiting too long in the other agent (waiting agent) control lanes. However, we find that the penalty has little effect on reducing traffic congestion and average waiting time. This is probably because the busy agent cannot hold an action for a long time in our simulation environment, which enables steady traffic streams in all directions. Therefore, the optimal control policy is to let the number of crossing vehicles be as large as possible. Besides, to avoid the action changes frequently, we set the minimum time interval for each agent. This setting may cause the control policy to fail to reach the desired results. Thus, we will aim to investigate how to set an optimal duration in traffic signal control based on DRL algorithms.

**Author Contributions:** Conceptualization, J.G. and Y.F.; Data curation, J.G. and Y.F.; Methodology, J.G. and Y.F.; Supervision Y.F., Z.S. and P.W.; Writing—original draft J.G.; Writing—review and editing J.G., Y.F., Z.S. and P.W. All authors have read and agreed to the published version of the manuscript.

**Funding:** This research is supported by the National Natural Science Foundation of China (61673253, 61901254) and the key support Projects of Shanghai Science and Technology Committee (16010500100).

**Conflicts of Interest:** The authors declare that there is no conflict of interest regarding the publication of this article.

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
