# Peer review of "Double Deep Q-Network with a Dual-Agent for Traffic Signal Control"

_applsci, doi:10.3390/app10051622_

Round 1

Reviewer 1 Report

It is a well sounding paper. However, there are some minor issues as follow.

Line 38, a citation regarding the congestion must be provided. Section 3.1, a formal proof that the proposed 2-agent architecture ensures the fixed phase sequence should be added to the paper. Section 5.1, line 251, is SUMO version 0.32.0 correct? Current version is 1.3.1. Line 290, it is not clearly visible that the proposed approach learns quicker, the learning curve in figures 9 and 10 seems to be quite similar. Source of the number of vehicles in the intersection should be explained, why is it 1800, 2520, 3240, 3600, 3960, 43200? Language must be corrected throughout the paper.

Author Response

    We would like to thank the reviewer for the careful reading and comments. Any revisions have been clearly highlighted in the revision with red. The reponse file is added in attachment with .docx format.

Reviewer 2 Report

Title : Double deep Q-Network with Dual-agent for Traffic Signal Control

Authors : Jianfeng Gu, Yong Fang, Zhichao Sheng and Peng Wen

Overview:

This paper proposes a double deep Q-Network with Dual-agent for Traffic Signal Control. Particularly, the two-agent solution performs in an alternate fashion. Moreover, a reward function is proposed in order to facilitate and accelerate training process. The methodology is evaluated on a four-line intersection using a simulated environment. Reported experiments show a clear improvement following three criteria: vehicle stop timing, average waiting and the number of arrived vehicles.

Hereafter some comments and questions concerning this paper:

This paper is well-written and can be read with ease. The proposed structure and the presented methods are sound, however, experiments are very limited. Some of the claims are hardly supported by the experiments that are actually reported.

Are production rates employed during simulation related to realistic scenarios? Are these rates representative of a real traffic management pattern on an intersection? (line 261)

In line 276, the authors mentioned that the simulation settings take into account driver reaction delays. What is the driver reaction time assumed? Can this timing be supported by bibliography?

Based on which criterion it is determined that 500 episodes are required to achieve an effective traffic signal control policy? (see line 270)

In Sec 5.2, a performance comparison during training is carried out and compared to a single agent approach. The single-agent approach was set so as to be stable in the intended scenario. This experiment certainly provides insights regarding the impact of using a two-agent approach. However, there is no proof or evidence that clearly supports the performance impact of the claimed contribution. There is not comparison with respect to other state of the art approaches.

How many training trials were conducted to verify the stability/gain of the training convergence?

In line 298, authors mentioned the use of a fixed timing method. Please provide the corresponding bibliography reference.

In Fig. 10, the authors inferred that the performance of the proposed algorithm is highly correlated to the traffic flow density employed during training. It is a pity that not further tests are reported concerning this issue. That seems to be also the case for [16].

As a claimed contribution, it is stated in the paper: Robustness of the proposed solution for efficient traffic management. However no proof or test seems to be reported concerning the robustness of the approach.

Minor issues:

Figures are not commented. Their labels barely present the figure but does not explain its content. Ex. Fig 1 and 2 What does the red-dashed region stand for?

Fig. 2 What are the operations representing the Agent?

ATSC is not defined: Adaptive Traffic Signal Control

L129 : … learns an effective control policy more fast. Faster

Table 2. Production rate stands for probability production rate, right?

Author Response

(The authors gave the same response as above.)
